# BigVGAN: A Universal Neural Vocoder with Large-Scale Training

**Sang-gil Lee**[1][*]    **Wei Ping** [2][†]

**Boris Ginsburg**[2]    **Bryan Catanzaro**[2]    **Sungroh Yoon**[1,3][†]

[1] Data Science & AI Lab, Seoul National University (SNU)

[2] NVIDIA

[3] AIIS, ASRI, INMC, ISRC, NSI, and Interdisciplinary Program in AI, SNU

```
tkdrlf9202@snu.ac.kr  wping@nvidia.com
bginsburg@nvidia.com  bcatanzaro@nvidia.com  sryoon@snu.ac.kr
```

## Abstract

Despite recent progress in generative adversarial network (GAN)-based vocoders, where the model generates raw waveform conditioned on acoustic features, it is challenging to synthesize high-fidelity audio for numerous speakers across various recording environments. In this work, we present BigVGAN, a universal vocoder that generalizes well for various out-of-distribution scenarios without fine-tuning. We introduce periodic activation function and anti-aliased representation into the GAN generator, which brings the desired inductive bias for audio synthesis and significantly improves audio quality. In addition, we train our GAN vocoder at the largest scale up to 112M parameters, which is unprecedented in the literature. We identify and address the failure modes in large-scale GAN training for audio, while maintaining high-fidelity output without over-regularization. Our BigVGAN, trained only on clean speech (LibriTTS), achieves the state-of-the-art performance for various zero-shot (out-of-distribution) conditions, including unseen speakers, languages, recording environments, singing voices, music, and instrumental audio. [1] We release our code and model at: https://github.com/NVIDIA/BigVGAN.

## 1 Introduction

Deep generative models have demonstrated noticeable successes for modeling raw audio. The successful methods include, autoregressive models (van den Oord et al., 2016; Mehri et al., 2017; Kalchbrenner et al., 2018), flow-based models (van den Oord et al., 2018; Ping et al., 2019; Prenger et al., 2019; Kim et al., 2019; Ping et al., 2020; Lee et al., 2020), GAN-based models (Donahue et al., 2019; Kumar et al., 2019; Bińkowski et al., 2020; Yamamoto et al., 2020; Kong et al., 2020), and diffusion models (Kong et al., 2021; Chen et al., 2021; Lee et al., 2022).

Among these methods, GAN-based vocoders (e.g., Kong et al., 2020) can generate high-fidelity raw audio conditioned on mel spectrogram, while synthesizing hundreds of times faster than real-time on a single GPU. However, existing GAN vocoders are confined to the settings with a moderate number of voices recorded in clean environment due to the limited model capacity. The audio quality can heavily degrade when the models are conditioned on mel spectrogram from unseen speakers in different recording environments. In practice, a *universal vocoder*, that can do zero-shot generation for out-of-distribution samples, is very valuable in many real-world applications, including text-to-speech with numerous speakers (Ping et al., 2018), neural voice cloning (Arik et al., 2018; Jia et al., 2018), voice conversion (Liu et al., 2018), speech-to-speech translation (Jia et al., 2019), and neural audio codec (Zeghidour et al., 2021). In these applications, the neural vocoder also needs to generalize well for audio recorded at various conditions.

---

[*]Work done during an internship at NVIDIA.

[†]Corresponding authors.

[1]Listen to audio samples from BigVGAN at: https://bigvgan-demo.github.io/.

Scaling up the model size for zero-shot performance is a noticeable trend in text generation (e.g., Brown et al., 2020) and image synthesis (e.g., Ramesh et al., 2021), but has not been explored in audio synthesis. Although likelihood-based models are found to be easier for scaling among others because of their simple training objective and stable optimization, we build our universal vocoder with large-scale GAN training, because GAN vocoder has the following advantages: *i*) In contrast to autoregressive or diffusion models, it is fully parallel and requires only one forward pass to generate high-dimensional waveform. *ii*) In contrast to flow-based models (Prenger et al., 2019), it does not enforce any architectural constraints (e.g., affine coupling layer) that maintain the bijection between latent and data. Such architectural constraints can limit model capacity given the same number of parameters (Ping et al., 2020).

In this work, we present *BigVGAN*, a *Big V*ocoding *GAN* that enables high-fidelity out-of-distribution (OOD) generation without fine-tuning. Specifically, we make the following contributions:

1. We introduce periodic activations into the generator, which provide the desired inductive bias for audio synthesis. Inspired by the methods proposed for other domains (Liu et al., 2020; Sitzmann et al., 2020), we demonstrate the noticeable success of periodic activations in audio synthesis.
2. We propose anti-aliased multi-periodicity composition (AMP) module for modeling complex audio waveform. AMP composes multiple signal components with learnable periodicities and uses low-pass filter to reduce the high-frequency artifacts.
3. We successfully scale BigVGAN up to 112M parameters by fixing the failure modes of large-scale GAN training without regularizing both generator and discriminator. The empirical insights are different from Brock et al. (2019) in image domain. For example, regularization methods (e.g., Miyato et al., 2018) introduce phase mismatch artifacts in audio synthesis.
4. We demonstrate that BigVGAN-base with 14M parameters outperforms the state-of-the-art neural vocoders with comparable size for both in-distribution and out-of-distribution samples. In particular, BigVGAN with 112M parameters outperforms the state-of-the-art models by a large margin for zero-shot generation at various OOD scenarios, including unseen speakers, novel languages, singing voices, music and instrumental audio in varied unseen recording environments.

We organize the rest of the paper as follows. We discuss related work in § 2 and present BigVGAN in § 3. We report empirical results in § 4 and conclude the paper in § 5.

## 2 RELATED WORK

Our work builds upon the state-of-the-art of GANs for image and audio synthesis. GAN was first proposed for image synthesis (Goodfellow et al., 2014). Since then, impressive results have been obtained through optimized architectures (e.g., Radford et al., 2016; Karras et al., 2021) or large scale training (e.g., Brock et al., 2019).

In audio synthesis, previous works focus on improving the discriminator architectures or adding new auxiliary training losses. MelGAN (Kumar et al., 2019) introduces the multi-scale discriminator (MSD) that uses average pooling to downsample the raw waveform at multiple scales and applies window-based discriminators at each scale separately. It also enforces the mapping between input mel spectrogram and generated waveform via an $\ell_1$ feature matching loss from discriminator. In contrast, GAN-TTS (Bińkowski et al., 2020) uses an ensemble of discriminators which operate on random windows of different sizes, and enforces the mapping between the conditioner and waveform adversarially using conditional discriminators. Parallel WaveGAN (Yamamoto et al., 2020) extends the single short-time Fourier transform (STFT) loss (Ping et al., 2019) to multi-resolution, and adds it as an auxiliary loss for GAN training. Yang et al. (2021) and Mustafa et al. (2021) further improve MelGAN by incorporating the multi-resolution STFT loss. HiFi-GAN (Kong et al., 2020) reuses the MSD from MelGAN, and introduces the multi-period discriminator (MPD) for high-fidelity synthesis. UnivNet (Jang et al., 2020; 2021) uses the multi-resolution discriminator (MRD) that takes the multi-resolution spectrograms as the input and can sharpen the spectral structure of synthesized waveform. In contrast, CARGAN (Morrison et al., 2022) incorporates the partial autoregression (Ping et al., 2020) into generator to improve the pitch and periodicity accuracy.

In this work, we focus on improving and scaling up the generator. We introduce the periodic inductive bias for audio synthesis and address the feature aliasing issues within the non-autoregressive generator architecture. Our architectural design has a connection with the latest results in time-series prediction (Liu et al., 2020), implicit neural representations (Sitzmann et al., 2020), and image synthesis (Karras et al., 2021). Note that, You et al. (2021) argues that different generator

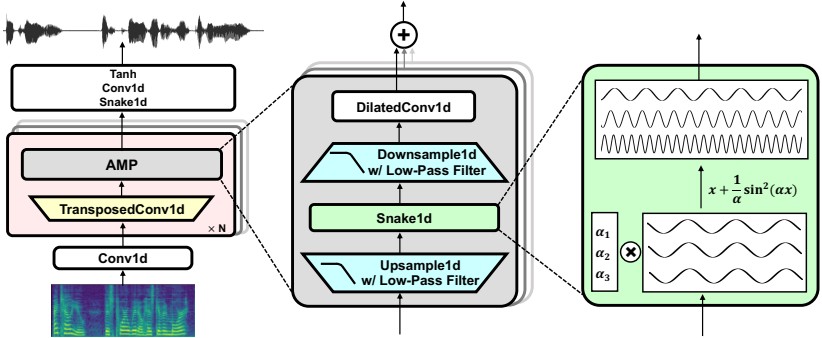

Figure 1: Schematic diagram of BigVGAN generator. The generator is composed of multiple blocks of transposed 1-D convolution followed by the proposed *anti-aliased multi-periodicity composition* (AMP) module. The AMP module adds features from multiple residual blocks with different channel-wise periodicities before dilated 1-D convolutions. It uses *Snake* function for providing periodic inductive bias, and low-pass filter for anti-aliasing purpose.

architectures can perform equally well for single-speaker neural vocoding. We demonstrate that improving generator architecture is crucial for universal neural vocoding in challenging conditions.

There are limited successes for universal neural vocoding due to the noticeable challenges. In previous work, WaveRNN has been applied for universal vocoding task (Lorenzo-Trueba et al., 2019; Paul et al., 2020). Jiao et al. (2021) builds the universal vocoder with flow-based model. GAN vocoder is found to be a good candidate recently (Jang et al., 2021).

## 3 METHOD

In this section, we introduce the preliminaries for GAN vocoder, then present the BigVGAN. See Figure 1 for an illustration and refer to the Appendix A for a detailed description of the architecture.

### 3.1 PRELIMINARIES OF GAN VOCODER

**Generator** The generator network takes mel spectrogram or other features as input and output the corresponding raw waveform. In previous studies, several generator architectures have been applied, including WaveNet (e.g., Yamamoto et al., 2020), or convolutional network that gradually upsamples the mel spectrogram to high-resolution waveform with a stack of residual blocks (e.g., Kumar et al., 2019; Kong et al., 2020). We choose the HiFi-GAN generator as the baseline architecture. We believe the proposed techniques are applicable to other generator architectures as well.

**Discriminator** The state-of-the-art GAN vocoders usually comprise several discriminators to guide the generator to synthesize coherent waveform while minimizing perceptual artifacts that are detectable by human ears. Importantly, each discriminator contains multiple sub-discriminators operating on different resolution windows of the waveform. For example, HiFi-GAN (Kong et al., 2020) applies two types of discriminators: *i*) the multi-period discriminator (MPD), where the 1-D signal is reshaped to 2-D representations with varied heights and widths to separately capture the multiple periodic structures though 2-D convolutions. *ii*) The multi-scale discriminator (MSD) (Kumar et al., 2019), where each sub-discriminator receives down-sampled 1-D signals at different frequency by average pooling in the time domain. Jang et al. (2020; 2021) propose to apply the discriminator on the time–frequency domain using the multi-resolution discriminator (MRD), which is composed of several sub-discriminators that operate on multiple 2-D linear spectrograms with different STFT resolutions. We also find that replacing MSD with MRD improves audio quality with reduced pitch and periodicity artifacts.

**Training objectives** Our training objective is similar as HiFi-GAN (Kong et al., 2020), with an exception of replacing MSD to MRD. It comprises the weighted sum of the least-square adversarial loss (Mao et al., 2017), the feature matching loss (Larsen et al., 2016), and the spectral $\ell_1$ regression loss on mel spectrogram. We leave the details of each loss and hyper-parameters in the Appendix B.

### 3.2 PERIODIC INDUCTIVE BIAS

The audio waveform is known to exhibit high periodicity and can be naturally represented as the composition of primitive periodic components (i.e., Fourier series under Dirichlet conditions). This

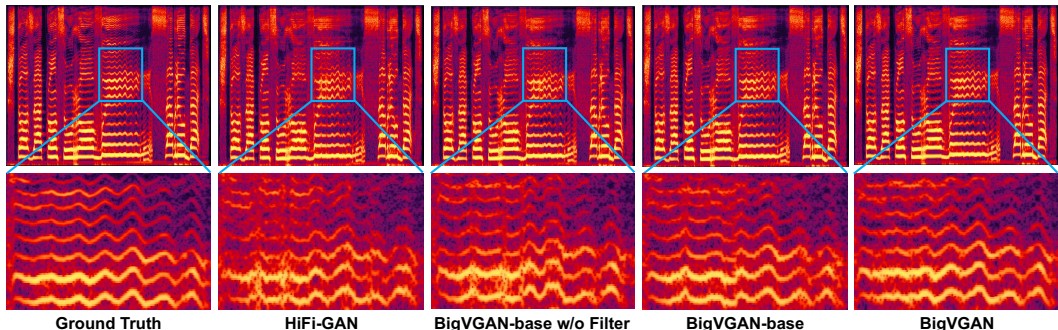

| Ground Truth | HiFi-GAN | BigVGAN-base w/o Filter | BigVGAN-base | BigVGAN |

Figure 2: Spectrogram visualization of a out-of-distribution sample (singing voice) from HiFi-GAN and BigVGAN trained on LibriTTS, with a zoomed in view of high-frequency harmonic components.

suggests that we need to provide the desired inductive bias to the generator architecture. However, the current non-autoregressive GAN vocoders (e.g., Kong et al., 2020) solely rely on layers of dilated convolutions to learn the required periodic components at different frequencies. Their activation functions (e.g., Leaky ReLU) can produce new details with necessary nonlinearities, but do not provide any periodic inductive bias. Furthermore, we identified that Leaky ReLU behaves poorly for *extrapolation* in waveform domain: although the model can generate high-quality speech signal in a seen recording environment at training, the performance degrades significantly for out-of-distribution scenarios such as unseen recording environments, non-speech vocalizations, and instrumental audio.

We introduce a proper inductive bias of periodicity to the generator by applying a recently proposed periodic activation called *Snake* function (Liu et al., 2020), defined as $f_\alpha(x) = x + \frac{1}{\alpha}\sin^2(\alpha x)$, where $\alpha$ is a trainable parameter that controls the frequency of the periodic component of the signal and larger $\alpha$ gives higher frequency. The use of $\sin^2(x)$ ensures monotonicity and renders it amenable to easy optimization. Liu et al. (2020) demonstrates this periodic activation exhibits an improved extrapolation capability for temperature and financial data prediction.

In BigVGAN, we use *Snake* activations $f_{\boldsymbol{\alpha}}(x)$ with channel-wise trainable parameters $\boldsymbol{\alpha} \in \mathbb{R}^h$ that define the periodic frequencies for each 1-D convolution channels. Taking this periodic functional form with learned frequency control, the convolutional module can naturally fit the raw waveform with multi-periodic components. We demonstrate that the proposed *Snake*-based generator is more robust for out-of-distribution audio samples unseen during training, indicating strong extrapolation capabilities in universal vocoding task. See Figure 2 and Appendix D for illustrative examples; BigVGAN-base w/o filter using snake activations is closer to ground-truth sample than HiFi-GAN.

### 3.3 ANTI-ALIASED REPRESENTATION

The *Snake* activations provide the required periodic inductive bias for modeling raw waveform, but it can produce arbitrary high frequency details for continuous-time signals that can not be represented by the discrete-time output of the network, [2] which can lead to aliasing artifacts. This side effect can be suppressed by applying a low-pass filter (e.g., Karras et al., 2021). The anti-aliased nonlinearity operates by upsampling the signal $2\times$ along time dimension, applying the *Snake* activation, then downsampling the signal by $2\times$, which is a common practice inspired by the Nyquist-Shannon sampling theorem (Shannon, 1949). Each upsampling and downsampling operation is accompanied by the low-pass filter using a windowed `sinc` filter with a Kaiser window (Oppenheim & Schafer, 2009). Refer to the Appendix A for details.

We apply this filtered *Snake* nonlinearity in every residual dilated convolution layers within the generator to obtain the anti-aliased representation of the discrete-time 1-D signals. The module is named as *anti-aliased multi-periodicity composition* (AMP). See Figure 1 for an illustration. We find that incorporating the filtered activation can reduce the high-frequency artifacts in the synthesized waveform; see BigVGAN-base w/o filter vs. BigVGAN-base (with filter) in Figure 2 as an illustration. We will demonstrate that it provides significant improvements in various objective and subjective evaluations. Note that we also explored anti-aliased upsampling layers, but this results in significant training instabilities and lead to early collapse for large models. See Appendix C for more details.

---

[2]One can think of the neural vocoder as a discrete-time function on the sampled continuous-time signals.

### 3.4 BigVGAN with Large Scale Training

In this subsection, we explore the limits of universal vocoding by scaling up the generator's model size to 112M parameters while maintaining the stability of GAN training and practical usability as a high-speed neural vocoder. We start with our improved generator using the comparable HiFi-GAN V1 configuration with 14M parameters (Kong et al., 2020), which is denoted as BigVGAN-base. We grow BigVGAN-base by increasing the number of upsampling blocks and convolution channels for each block. The BigVGAN-base upsamples the signal by $256\times$ using 4 upsampling blocks with the ratio of $[8, 8, 2, 2]$. Each upsampling block is accompanied by multiple residual layers with dilated convolutions, i.e., the AMP module. We further divides the $256\times$ upsampling into 6 blocks $[4, 4, 2, 2, 2, 2]$ for more fine-grained feature refinement. In addition, we increase the number of channels of AMP module (analogous to MRF in HiFi-GAN) from 512 to 1536. We denote the model with 1536 channels and 112M parameters as BigVGAN.

We found that the default learning rate of $2 \times 10^{-4}$ used in HiFi-GAN causes an early training collapse for BigVGAN training, where the losses from the discriminator submodules immediately converge to zero after several thousands of iterations. Halving the learning rate to $1 \times 10^{-4}$ was able to reduce such failures. We also found that large batch size is helpful to reduce mode collapse at training (Brock et al., 2019). We only double the batch size from the usual 16 to 32 for a good trade-off between training efficiency and stability, as neural vocoders can require millions of steps to converge. Note that this recommended batch size is still much smaller than the one for image synthesis (e.g., 2048) (Brock et al., 2019), because neural vocoding has strong conditional information.

Even with the aforementioned changes, the large BigVGAN can still be prone to collapse early in training. We track the gradient norm of each modules during training and identify that the anti-aliased nonlinearity significantly amplified the gradient norm of MPD. Consequently, BigVGAN generator receives a diverging gradient early in training, leading to instabilities and potential collapse. We visualize the norm of gradient for each modules in Figure 4 at Appendix C. We alleviate the issue by clipping the global norm of the gradient to $10^3$, which is close to the average gradient norm of the 112M BigVGAN generator. This gradient clipping prevents the early training collapse of the generator. Note that, gradient clipping was found ineffective to alleviate training instability for image synthesis (see Appendix H in Brock et al. (2019)), but it is very effective in our endeavors.

In addition to above efforts, we have explored other directions, including various ways to improve the model architecture, spectral normalization (Miyato et al., 2018) to stabilize GAN training, which is crucial for large-scale GAN training in image domain, and data augmentation to improve model generalization. Unfortunately, all these trials resulted in worse perceptual quality in our study. The details can be found in the Appendix C. We hope these practical lessons that we have learned would be useful to future research endeavors.

## 4 Results

We conduct a comprehensive evaluation of BigVGAN for both in-distribution and out-of-distribution scenarios. We train BigVGAN and all baseline models on the full LibriTTS dataset.

### 4.1 Training Data

We use LibriTTS (Zen et al., 2019) dataset with the original sampling rate of 24 kHz for training. Unlike previous studies which only adopted a subset (`train-clean-100` or `train-clean-360`) recorded in a clean environment (Jang et al., 2020; 2021; AlBadawy et al., 2022), we use all training data including the subset from diverse recording environments (`train-full` = `train-clean-100` + `train-clean-360` + `train-other-500`), which is unprecedented in the literature. We find that the diversity of the training data is important to achieve the goal towards universal neural vocoding using BigVGAN. [3] For OOD experiments, we resample the audio to 24 kHz if necessary using `kaiser-best` algorithm provided by `librosa` package.

Conventional STFT parameters are engineered to have a limited frequency band [0, 8] kHz by cutting off the high frequency details for easier modeling. On a contrary, we train all models (including the baselines) using a frequency range [0, 12] kHz and a 100-band log-mel spectrogram, which is also used in a recent study towards universal vocoding (Jang et al., 2021). We set other STFT parameters as in previous work (Kong et al., 2020), with 1024 FFT size, 1024 Hann window, and 256 hop size.

---

[3]The ablation results on training data diversity can be found in Table 5.

Table 1: Model footprint and synthesis speed for 24 kHz audio measured on an NVIDIA RTX 8000 GPU.

| Method | WaveGlow | WaveFlow | HiFi-GAN (V1) | BigVGAN-base | w/o filter | BigVGAN |
|---|---|---|---|---|---|---|
| Params (M) | 99.43 | 22.58 | 14.01 | 14.01 | 14.01 | 112.4 |
| Syn. speed | 31.87× | 19.59× | 93.75× | 70.18× | 75.83× | 44.72× |

## 4.2 MODELS

We train all BigVGAN models including the ablation models and the baseline HiFi-GAN using our training configuration for 1M steps. We use the batch size of 32, a segment size of 8,192, and a initial learning rate of $1 \times 10^{-4}$. All other configurations including optimizer, learning rate scheduler, and scalar weights of the loss terms follow the official open-source implementation of HiFi-GAN (Kong et al., 2020) without modification, with an exception that BigVGAN replaces MSD by MRD for the discriminator. All models are trained using NVIDIA DGX-1 with 8 V100 GPUs. Refer to Table 6 in the Appendix A for detailed hyperparameters.

We include a comparison with SC-WaveRNN (Paul et al., 2020), a state-of-the-art autoregressive universal neural vocoder based on WaveRNN (Kalchbrenner et al., 2018), using the official implementation. We also include two popular flow-based models: WaveGlow (Prenger et al., 2019) and WaveFlow (Ping et al., 2020), using their official implementation. For out-of-distribution test, we include the unofficial open-source implementation of UnivNet-c32 (Jang et al., 2021), [4] which uses `train-clean-360` subset for training and is reported to outperform HiFi-GAN under the same training configurations. See appendix E for more details.

Table 1 summarizes the synthesis speed of flow-based and GAN vocoders for generating 24 kHz audio. We omit SC-WaveRNN as it is much slower. BigVGAN-base with 14M parameters can synthesize the audio 70.18× faster than real time, which is relatively slower than HiFi-GAN as the filtered *Snake* function requires more computation. HiFi-GAN and BigVGAN are faster than flow-based models, because they are fully parallel (WaveFlow has partial autoregression) and have much fewer layers (WaveGlow has 96 layers). Our BigVGAN with 112M parameters can synthesize the audio 44.72 × faster than real-time and keeps the promise as a high-speed neural vocoder.

## 4.3 EVALUATION METRICS

The objective metrics we collected are designed to measure varied types of distance between the ground-truth audio and the generated sample. We provide 5 different metrics: 1) Multi-resolution STFT (M-STFT) (Yamamoto et al., 2020) which measures the spectral distance across multiple resolutions. [5] 2) Perceptual evaluation of speech quality (PESQ) (Rix et al., 2001), a widely adopted automated assessment of voice quality. [6] 3) Mel-cepstral distortion (MCD) (Kubichek, 1993) with dynamic time warping which measures the difference between mel cepstra. [7] 4) Periodicity error, and 5) F1 score of voiced/unvoiced classification (V/UV F1) which are considered as major artifacts from non-autoregressive GAN vocoders (Morrison et al., 2022). [8]

The conventional 5-scale mean opinion score (MOS) is insufficient for the subjective evaluation of universal vocoder, because the metric needs to differentiate the utterances from diverse speaker identities recorded in various environments. For example, the model may always output some very natural "average" voices, which is not preferred but can still be highly rated by human workers in MOS evaluation. As a result, we also perform the 5-scale similarity mean opinion score (SMOS) evaluation, where the participant is asked to give the score of similarity for the pair of audio after listening to ground-truth audio and the sample from the model side-by-side. SMOS provides an improved way of assessing how close the given sample is to the ground-truth, where the ground-truth recordings can have diverse speaker identities, contains unseen languages for the listeners, and be recorded in various acoustic environments. SMOS is also directly applicable to non-speech samples, e.g., music. We did MOS and SMOS evaluation on Mechanical Turk. More details can be found in Appendix G.

---

[4] `https://github.com/mindslab-ai/univnet`. Note there is no official open-source code.

[5] We used an open-source implementation from `Auraloss` (Steinmetz & Reiss, 2020).

[6] We used a 16,000Hz wide-band version from `https://github.com/ludlows/python-pesq`.

[7] We used an open-source implementation from `https://github.com/ttslr/python-MCD`.

[8] We used the periodicity error and V/UV F1 score code provided by CARGAN (Morrison et al., 2022).

Table 2: Objective and subjective quality metrics of BigVGAN evaluated on LibriTTS. Objective results are obtained from `dev` sets, and subjective evaluations with 5-scale mean opinion score (MOS) and similarity mean opinion score (SMOS) with 95% confidence interval (CI) are obtained from `test` sets.

| LibriTTS | M-STFT(↓) | PESQ(↑) | MCD(↓) | Periodicity(↓) | V/UV F1(↑) | MOS(↑) | SMOS(↑) |
|---|---|---|---|---|---|---|---|
| Ground Truth | - | - | - | - | - | 4.40±0.06 | 4.44±0.06 |
| SC-WaveRNN | 2.2358 | 1.701 | 1.8854 | 0.3044 | 0.8144 | 3.20±0.11 | 3.29±0.10 |
| WaveGlow-256 | 1.3099 | 3.138 | 2.3591 | 0.1485 | 0.9378 | 3.84±0.10 | 3.87±0.10 |
| WaveFlow-128 | 1.1120 | 3.027 | 1.2455 | 0.1416 | 0.9410 | 3.85±0.10 | 3.89±0.10 |
| HiFi-GAN (V1) | 1.0017 | 2.947 | 0.6603 | 0.1565 | 0.9300 | 4.08±0.09 | 4.15±0.09 |
| BigVGAN-base | 0.8788 | 3.519 | 0.4564 | 0.1287 | 0.9459 | 4.10±0.09 | 4.20±0.08 |
| BigVGAN | **0.7997** | **4.027** | **0.3745** | **0.1018** | **0.9598** | **4.11±0.09** | **4.26±0.08** |

Table 3: The 5-scale SMOS results with 95% CI evaluated on unseen languages with different types of noise in unseen recording environments. †: pretrained weight obtained from an open-source repository which used `train-clean-360` subset for training.

| Recording env. Language | Clean Jv,Km,Ne,Su | Noisy (sim) Es,Fr,It,Pt | Noisy (real) Ko |
|---|---|---|---|
| Ground Truth | 4.58±0.05 | 4.36±0.05 | 4.56±0.05 |
| UnivNet-c32† | 4.35±0.07 | 3.95±0.09 | 4.18±0.08 |
| HiFi-GAN (V1) | 4.39±0.07 | 4.13±0.08 | 4.21±0.08 |
| BigVGAN-base | 4.38±0.07 | 4.21±0.07 | 4.36±0.07 |
| BigVGAN | **4.41±0.07** | **4.26±0.07** | **4.38±0.07** |

## 4.4 LibriTTS Results

We report the performance of BigVGAN and the baseline models evaluated on LibriTTS using above objective and subjective metrics. We perform objective evaluations on `dev-clean` and `dev-other` altogether, and conduct subjective evaluations on the combined `test-clean` and `test-other`. The `dev` and `test` splits of LibriTTS contains unseen speakers during training, but the recording environments are covered in the train split.

Table 2 shows the in-distribution test results on LibriTTS. Baseline models other than HiFi-GAN performs significantly worse. This indicates that GAN vocoder is the state-of-the-art for universal neural vocoding. BigVGAN significantly improves all objective metrics. In particular, BigVGAN-base exhibits consistently improved objective scores over HiFi-GAN (V1) with the same amount of paramters, suggesting that it has better periodic inductive bias for waveform data.

HiFi-GAN (V1), BigVGAN-base, and BigVGAN perform comparably well in terms of MOS without listening to the ground-truth audio side-by-side. When the listeners can compare the model sample with ground truth audio side-by-side, BigVGAN-base measurably outperforms HiFi-GAN (V1) in terms of SMOS (+0.05), and the 112M BigVGAN outperforms HiFi-GAN by a clear margin in terms of SMOS (+0.11) because it has high model capacity to further leverage the diverse training data for better quality.

In Appendix E, we also report the additional results including UnivNet (Jang et al., 2021) and the ablation models of BigVGAN-base on LibriTTS `dev` sets, unseen `VCTK` (Yamagishi et al., 2019), and `LJSpeech` (Ito, 2017) data.

## 4.5 Unseen Languages and Varied Recording Environments

In this subsection, we assess the universal vocoding capability of BigVGAN by measuring its zero-shot performance for various unseen languages with varied types of the recording environments in the unseen dataset. Based on the results in Table 2, we only include GAN-based vocoders as the state-of-the-art baseline. We gather three classes of a publicly available multi-language dataset categorized by the type of noise from the recording environment.

- A collection of under-resourced languages recorded in a noiseless studio environment (Sodimana et al., 2018): Javanese, Khmer, Nepali, and Sundanese. We use randomly selected 50 audio clips with equal balance across languages from the combined dataset.

Table 4: The 5-scale SMOS results with 95% CI evaluated on out-of-distribution samples from MUSDB18-HQ. †: pretrained model from an open-source repository which used `train-clean-360` subset for training.

| Method | Vocal | Drums | Bass | Others | Mixture | Average |
|---|---|---|---|---|---|---|
| Ground Truth | 4.58±0.05 | 4.57±0.05 | 4.52±0.05 | 4.61±0.05 | 4.56±0.05 | 4.57±0.02 |
| UnivNet-c32† | 4.22±0.09 | 4.23±0.09 | 3.90±0.11 | 3.80±0.13 | 3.80±0.12 | 3.99±0.05 |
| HiFi-GAN (V1) | 4.26±0.08 | 4.37±0.08 | 3.95±0.11 | 3.92±0.12 | 3.91±0.11 | 4.08±0.05 |
| BigVGAN-base | 4.36±0.08 | 4.39±0.07 | **4.00 ±0.11** | 4.14±0.09 | 4.11±0.10 | 4.20±0.04 |
| w/o filter | 4.30±0.08 | 4.32±0.07 | 3.95±0.11 | 4.05±0.10 | 4.11±0.10 | 4.15±0.04 |
| w/o filter & snake | 4.31±0.08 | 4.32±0.07 | 3.94±0.11 | 4.01±0.11 | 4.02±0.10 | 4.12±0.04 |
| BigVGAN | **4.37±0.08** | **4.41±0.07** | **4.00±0.10** | **4.25±0.09** | **4.26±0.08** | **4.26±0.04** |

- The Multilingual TEDx Corpus (Salesky et al., 2021): contains a collection of TEDx talks in Spanish, French, Italian, and Portuguese. We use randomly selected 50 audio clips with equal balance across languages from the IWSLT'21 test set. We simulate the unseen recording environment setup by adding a random environmental noise from MS-SNSD (Reddy et al., 2019), such as airport, cafe, babble, etc.

- Deeply Korean read speech corpus (Deeply, 2021): contains short speech audio clips in Korean, recorded in three types of recording environments (anechoic chamber, studio apartment, and dance studio) using a smartphone. We use randomly selected 50 audio clips where 25 clips are from the studio apartment, and the remaining 25 clips are from the dance studio. The collected audio clips contain a significant amount of noise and artifacts from real-world recording environments, such as reverb, echo, and static background noise.

Table 3 summarizes the SMOS results from three different types of unseen dataset. We only did SMOS evaluations, because the datasets have unseen languages for human listeners and it is hard to determine the quality without side-by-side comparison with the ground-truth recordings. For clean under-resourced language dataset, the performance gap between models is not substantially large. This indicates that the universal vocoder trained on the entire LibriTTS training set is robust to unseen languages under clean recording environments. For both types of unseen recording environment (simulated or real-world), BigVGAN outperforms the baseline models by a large margin. The small capacity BigVGAN-base also shows improvements compared to the baseline with statistical significance (p-value $< 0.05$ from the Wilcoxon signed-rank test). This suggests that BigVGAN is significantly more robust to the unseen recording environments thanks to the improved generator design with the AMP module. In Appendix F, we further demonstrate that BigVGAN is the most linguistically accurate universal vocoder in terms of character error rate (CER) on multiple languages.

We test the open-source implementation of UnivNet (Jang et al., 2021) with the pretrained checkpoint which is trained on `train-clean-360` subset. Contrary to the report from Jang et al. (2021) that UnivNet-c32 outperformed HiFi-GAN (Kong et al., 2020), we find that the unmodified HiFi-GAN trained on the entire LibriTTS dataset is able to match or outperform UnivNet-c32. We also train UnivNet-c32 on LibriTTS `train-full` and find that it is not benefited from larger training data. See Appendix E for detailed analysis.

### 4.6 OUT-OF-DISTRIBUTION ROBUSTNESS

In this subsection, we test BigVGAN's robustness and extrapolation capability by measuring zero-shot performance on out-of-distribution data. We conduct the SMOS experiment using MUSDB18-HQ (Rafii et al., 2019), a multi-track music audio dataset which contains vocal, drums, bass, other instruments, and the original mixture. The test set contains 50 songs with 5 tracks. We gather the mid-song clip with the duration of 10 seconds for each track and song.

Table 4 shows the SMOS results from the 5 tracks and their average from the MUSDB18-HQ test set. BigVGAN models demonstrate a substantially improved zero-shot generation performance with wider frequency band coverage, whereas baseline models fail to generate audio outside the limited frequency range and suffer from severe distortion. The improvements are most profound for singing voice (vocal), instrumental audio (others) and the full mixture of the song (mixture), whereas the improvements from drums and bass are less significant.

Table 5: Ablation results on training data diversity using 112M BigVGAN model, evaluated on LibriTTS. Objective results are obtained from `dev-other` and subjective evaluation with 5-scale SMOS with 95% confidence interval (CI) is obtained from `test-other`.

| Training data | M-STFT(↓) | PESQ(↑) | MCD(↓) | Periodicity(↓) | V/UV F1(↑) | SMOS(↑) |
|---|---|---|---|---|---|---|
| Ground Truth | - | - | - | - | - | 4.55±0.05 |
| `train-full` | **0.8197** | **4.001** | **0.4097** | **0.1023** | **0.9586** | **4.38±0.07** |
| `train-clean-360` | 0.8429 | 3.847 | 0.4232 | 0.1149 | 0.9521 | 4.31±0.08 |
| `VCTK` | 0.8747 | 3.818 | 0.5921 | 0.1215 | 0.9490 | 4.27±0.08 |

We also experiment with audios obtained from YouTube videos from real-world recording environments. BigVGAN also exhibits robustness to various types of out-of-distribution signals such as laughter. We provide audio samples to our demo page. [9]

### 4.7 ABLATION STUDY

**Model architecture** To measure the effectiveness of the BigVGAN generator, we include SMOS test for the ablation models of BigVGAN-base on MUSDB18-HQ data. Table 4 shows that the ablation models exhibit clear degradation on various scenarios such as instrumental audio (others, mixture). From the average SMOS ratings, 1) disabling the anti-aliasing filter for *Snake* activation performs worse than BigVGAN-base and 2) removing both the filter and *Snake* activation (i.e., vanilla HiFi-GAN trained with MRD replacing MSD) is even worse than the *Snake*-only ablation model, both with statistical significance (p-value < 0.01 from the Wilcoxon signed-rank test). This indicates that Leaky ReLU is not robust enough to extrapolate beyond the learned frequency range and the aliasing artifacts degrade the audio quality in challenging setups. The results show that BigVGAN generator demonstrates strong robustness and extrapolation capability to out-of-distribution scenarios because of the seamless integration of periodic inductive bias and anti-aliased feature representation. See Appendix D for the visualization of anti-aliasing effect in BigVGAN.

**Big model** We compare HiFi-GAN and BigVGAN both with the largest 112M parameters. We train the 112M HiFi-GAN with the same training setting as BigVGAN. We conduct a pairwise test between the two models on the `mixture` test set of MUSDB18-HQ which is challenging out-of-distribution data. We ask the participants to select a better sounding audio between the samples from the two models. The test shows that 58 % of the ratings voted to BigVGAN over the large HiFi-GAN and the quality of BigVGAN is greater than the large HiFi-GAN with statistical significance (p-value < 0.01 from the Wilcoxon signed-rank test). The results further validate the architectural advantage of BigVGAN in large-scale setting.

**Large Training data** To verify the importance of using large-scale training data, we trained our BigVGAN using less diverse, clean speech-only dataset with the same training configuration for 1M steps: 1) `train-clean-360` subset of LibriTTS, or 2) VCTK dataset. Table 5 shows that training BigVGAN on less diverse data shows degradation in both objective metrics and the subjective SMOS on the LibriTTS evaluation sets. The result verifies the importance of using diverse training data and demonstrates the effectiveness of BigVGAN on large-scale datasets.

## 5 CONCLUSIONS

This study explores the limits of universal neural vocoding with an unprecedented scale of the data, model, and evaluations. We analyze the performance with various automatic and human evaluations across diverse scenarios including unseen speakers, languages, recording environments and out-of-distribution data. We present BigVGAN with an improved generator architecture by introducing anti-aliased periodic activation function with learned frequency control, which injects the desired inductive bias for waveform generation. Based on the improved generator, we demonstrate the largest GAN vocoder with strong zero-shot performance under various OOD conditions, including unseen recording environments, singing voice, and instrumental audio. We believe that BigVGAN, combined with practical lessons learned from the large scale training, will inspire future endeavors for universal vocoding and improve the state-of-the-art results for real-world applications, including voice cloning, voice conversion, speech translation, and audio codec.

---

[9]https://bigvgan-demo.github.io

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

# APPENDIX

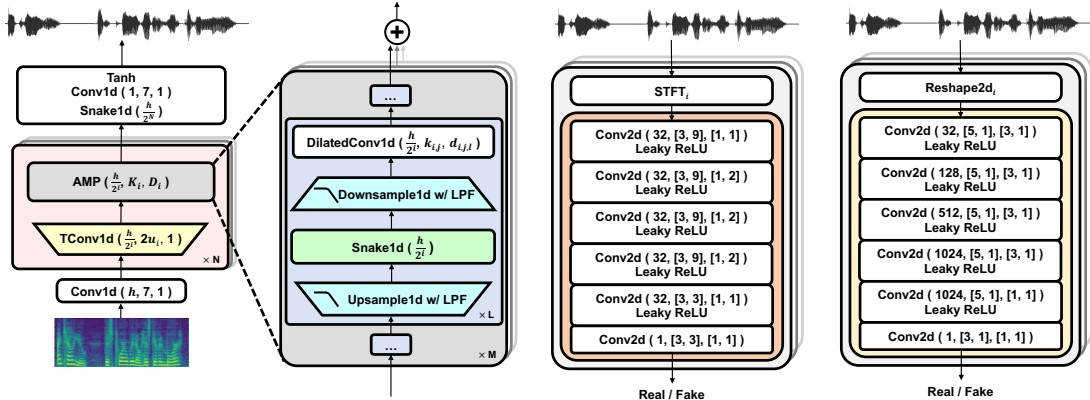

Figure 3: Detailed diagram of BigVGAN generator and discriminator architectures. Left: generator architecture, where values in parentheses denote (output channel, kernel width, dilation rate) respectively. Right: discriminator architectures (MRD in orange and MPD in yellow), where values in parentheses denote (output channel, [kernel width, kernel height], [stride width, stride height]) respectively.

Table 6: Hyperparameters of BigVGAN generators and discriminators.

| | Generator | | Discriminator | |
|---|---|---|---|---|
| | BigVGAN-base | BigVGAN | MRD & MPD | |
| $h$ | 512 | 1536 | $\texttt{n\_fft}_i$ | [1024, 2048, 512] |
| $u_i$ | [8, 8, 2, 2] | [4, 4, 2, 2, 2, 2] | $\texttt{hop\_length}_i$ | [120, 240, 50] |
| $K_i$ | [3, 3, 3, 7, 7, 7, 11, 11, 11] | [3, 3, 3, 7, 7, 7, 11, 11, 11] | $\texttt{win\_length}_i$ | [600, 1200, 240] |
| $D_i$ | [[1, 1], [3, 1], [5, 1]] × 3 | [[1, 1], [3, 1], [5, 1]] × 3 | $\texttt{Reshape2d}_i\,(p_i)$ | [2, 3, 5, 7, 11] |

## A    ARCHITECTURAL DETAILS

In this section, we present a detailed description of the BigVGAN architecture. Refer to Figure 3 for illustrative details. BigVGAN uses the similar generator architecture presented in HiFi-GAN (Kong et al., 2020). The generator takes a mel spectrogram as input and synthesizes a corresponding waveform as output. The generation process starts with a single layer of $1D$ convolution using a channel width of $h$ and a kernel size of 7 without dilation (denoted as 1 in the Figure 3). The hierarchical generation comprises $N$ number of upsampling blocks. The $i$-th upsampling block ($i = \{1, \ldots, N\}$) starts with a transposed $1D$ convolution using half the number of channels of the preceding block and an upsampling rate of $u_i$. The upsampled feature is followed by $M$ number of AMP residual blocks, where each AMP block uses different kernel sizes for a stack of dilated $1D$ convolutions defined as $k_{i,j}(j = \{1, \ldots, M\})$. The $j$-th AMP block contains $L$ number of the anti-aliased periodic activation and the dilated $1D$ convolution using a dilatation rate of $d_{i,j,l}(l = \{1, \ldots, L\})$. Refer to the Table 6 for the hyperparameters of the BigVGAN generators.

Our design of the low-pass filter is similar to StyleGAN3 (Karras et al., 2021). We use a cutoff frequency of $\frac{s}{2m}$, where $m = 2$ is a up- and down-sampling ratio and $s$ is a sampling rate (e.g., width) of the signal. The Kaiser window uses a window length of $n = 6 \cdot m$ and a shape parameter $\beta$ is approximated by $0.1102 \cdot (A - 8.7)$, where a maximum attenuation $A$ is approximated by $A = 2.285 \cdot (\frac{n}{2} - 1) \cdot \pi \cdot 4f_h + 7.95$ (Oppenheim & Schafer, 2009) with a transition band half-width $f_h = \frac{0.6}{m}$. The low-pass filter is applied as the kernel in $1D$ convolution for downsampling, and as the kernel in transposed $1D$ convolution for upsampling.

The BigVGAN discriminator comprises two submodules: MRD and MPD. Each module is composed of multiple subdiscriminators using a stack of $2D$ convolutions as in Figure 3. MRD converts the

input $1D$ waveform to its $2D$ linear spectrogram using STFT with different parameters ([n_fft, hop_length, win_length]). MPD converts the input $1D$ waveform with length $T$ to its $2D$ representation by reshaping and reflection padding (Reshape2d) with different width ($p_i$) and height ($\frac{T}{p_i}$). Refer to the Table 6 for the MRD and MPD hyperparameters. We used the same MRD and MPD hyperparameters for training all BigVGAN generators.

## B TRAINING OBJECTIVE DETAILS

We apply the training objective formulation and its hyperparameters described in (Kong et al., 2020) without modification, with an exception that BigVGAN applies MRD replacing MSD as the discriminator submodule. Concretely, we apply the following objectives $\mathcal{L}_G$ for generator and $\mathcal{L}_D$ for discriminator, respectively:

$$\mathcal{L}_G = \sum_{k=1}^{K}\left[\mathcal{L}_{adv}(G;D_k)+\lambda_{fm}\mathcal{L}_{fm}(G;D_k)\right]+\lambda_{mel}\mathcal{L}_{mel}(G), \quad \mathcal{L}_D = \sum_{k=1}^{K}\left[\mathcal{L}_{adv}(D_k;G)\right], \quad (1)$$

where $D_k$ denotes the $k$-th MPD or MRD discriminator submodules. $\mathcal{L}_{adv}$ uses the least-square GAN (Mao et al., 2017) as follows:

$$\mathcal{L}_{adv}(G;D_k) = \mathbb{E}_s\left[(D_k(G(s))-1)^2\right], \quad \mathcal{L}_{adv}(D_k;G) = \mathbb{E}_{(x,s)}\left[(D_k(x)-1)^2+(D_k(G(s)))^2\right], \quad (2)$$

where $x$ is the ground-truth waveform, and $s$ is the input mel spectrogram. The feature matching loss $\mathcal{L}_{fm}$ (Larsen et al., 2016; Kumar et al., 2019) minimizes the $\ell_1$ distance for every intermediate features from the discriminator layers:

$$\mathcal{L}_{fm}(G;D_k) = \mathbb{E}_{(x,s)}\left[\sum_{i=1}^{T}\frac{1}{N}||D_k^i(x)-D_k^i(G(s))||_1\right], \quad (3)$$

where $T$ is the number of layers of the sub-discriminator $D_k$. The generator loss $\mathcal{L}_G$ also has the spectral $\ell_1$ regression loss between the mel spectrogram of the synthesized waveform and the corresponding ground-truth:

$$\mathcal{L}_{mel}(G) = \mathbb{E}_{(x,s)}\left[||\phi(x)-\phi(G(s))||_1\right], \quad (4)$$

where $\phi$ is the STFT function that converts the waveform into the mel spectrogram. We used the scalar weights $\lambda_{fm} = 2$ and $\lambda_{mel} = 45$ identically as (Kong et al., 2020).

## C PRACTICAL LESSONS FOR LARGE-SCALE GAN TRAINING ON AUDIO

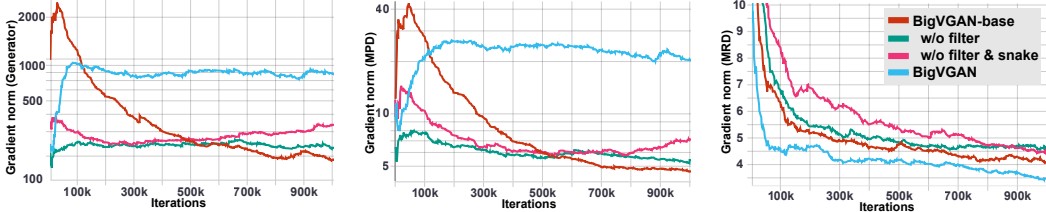

Figure 4: Visualization of gradient norm for different modules from BigVGAN training. Left: Gradient norm from the generator. Mid: Gradient from the MPD module. Right: Gradient norm from the MRD module. For BigVGAN-base, the gradient norm significantly increases at early training without clipping. For BigVGAN, the gradient will explode without clipping.

In this section, we document additional directions that we have explored for improving the model architecture and large scale training, but resulted in worse perceptual quality from our preliminary study. We do not make conclusive claims based on these observations because the methods we explored here may have been ineffective specific to our BigVGAN settings. Nevertheless, we believe that reporting negative results can be helpful to future research endeavors (Brock et al., 2019).

**Anti-aliased upsampling layers**    Our BigVGAN uses anti-aliased activation for AMP modules. We also explored replacing the transposed convolution-based upsampling layers, which are known to contain checkerboard aliasing (Odena et al., 2016), to the anti-aliased alternatives with different low-pass filter hyperparameters. However, this introduced significant instabilities during training, leading to early collapse even with the aforementioned stabilization. We also tried out nearest-neighbor upsampling which is reported to have less artifacts for audio synthesis (Pons et al., 2021), but it also resulted in the early collapse.

**Periodic discriminators**    Inspired by the improvements with the periodic activation function to the generator, we experimented on discriminators with *Snake* function. However, it degraded the quality with the diverging feature matching loss (Kong et al., 2020) from the discriminators. We conjecture that the periodic activation to the discriminator is not stable enough to improve the gradient from the feature matching loss.

**Spectral normalization**    Spectral normalization (Miyato et al., 2018) is a widely adopted method to stabilize GAN training in image domain. We tried applying spectral normalization to all discriminator submodules and found that it can stabilize the training without the early divergence of the generator. However, it suffered from a significant degradation with the excessive amount of phase mismatch artifacts, similar to the findings in the previous work (Kumar et al., 2019). We found that the gradient from MPD is over-regularized and the generator start to solely rely on the mel regression loss. Because MPD is repeatedly found to be a key component for high-quality audio synthesis (Kong et al., 2020), regularizing MPD leads to worse result.

**Larger discriminators**    We hypothesized that the enlarged generator can be slower to learn, thereby trivializing the discriminator in the early training (loss converge to zero). We tried to balance the training by enlarging the discriminators, such as employing more MPD sub-discriminators or enlarging the channel width of MPD and MRD modules. The large discriminator partially alleviated the early collapse, but the audio quality degraded in most cases, and showed no clear improvements even with our best configuration.

**Even larger generators**    We experimented with deeper model with 8 upsampling blocks with ratio of $[2, 2, 2, 2, 2, 2, 2, 2]$. However, it exhibited high-frequency rattling artifacts and degraded the quality. We conjecture that generating fine-grained high-frequency details from the early upsampling blocks can be arbitrary (Karras et al., 2021) and unstable when using the filtered periodic nonlinearity. We also tried to further increase the number of convolution channels to $2048$, but it suffered from early collapse.

**Data augmentation**    Data augmentation is one of major methods that improve model generalization, which is also repeatedly found to be valuable in GAN literature (Karras et al., 2020). We explored augmenting the data by applying SpecAugment (Park et al., 2019) to the input mel spectrogram. However, SpecAugment resulted in over-smoothing artifacts in waveform, because it enforces the model to map multiple distorted mel spectrograms to the same waveform, which counters the upsampling process in generative models. We also tried applying mixup-like (Zhang et al., 2018) approach to the waveform by using a convex combination of two audio samples and its corresponding mel spectrogram. However, this occasionally resulted in a mixed voice of two speaker identities during a single-speaker inference without noticeable improvement in perceptual quality.

**Positional encoding**    Inspired by using partial autoregression (Ping et al., 2020; Morrison et al., 2022) to provide inductive bias of the cumulative sum relationship of pitch and phase (Morrison et al., 2022), we explored whether we can provide an approximate inductive bias in a non-autoregressive manner by injecting a sinusoidal positional encoding (Vaswani et al., 2017) to the generator. However, the generator with positional encoding is only exposed to the fixed audio segment at training ($\sim$0.3 seconds with 8,192 time-steps) and unable to extrapolate unknown and significantly longer sequence at inference (e.g., several seconds).

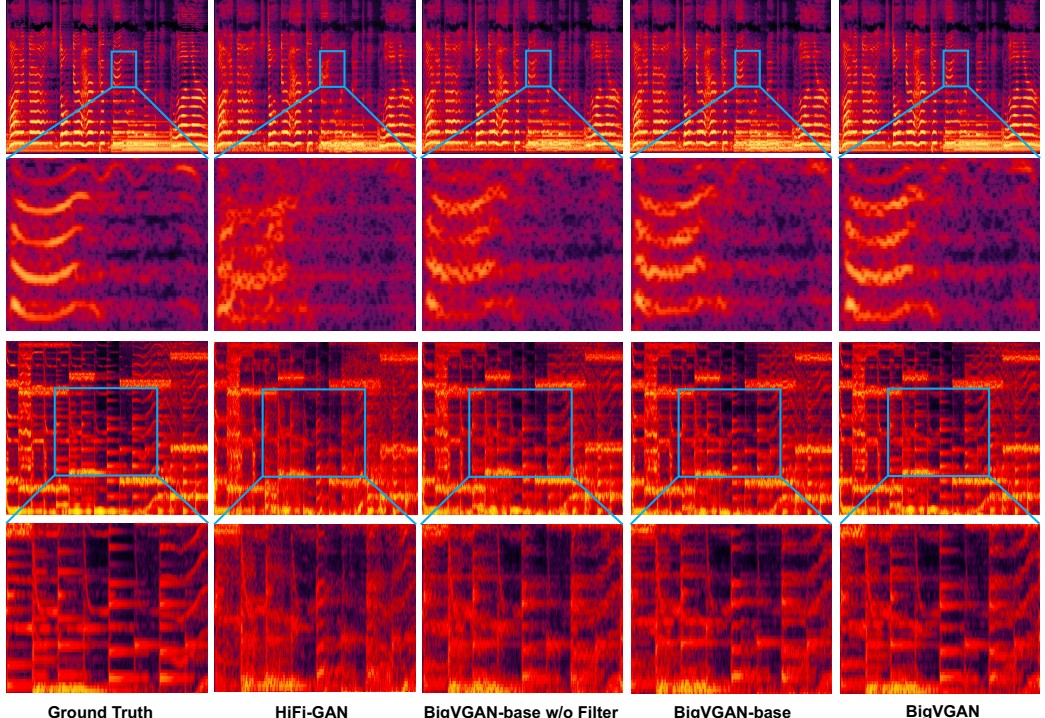

Figure 5: Spectrogram visualization of out-of-distribution samples from HiFi-GAN and BigVGAN models trained on LibriTTS, with a zoomed in view of harmonic components. Top: a singing voice. Bottom: an instrumental audio.

## D VISUALIZATION

In this section, we present visual examples to analyze the improvements from the methodologies used in BigVGAN. We use HiFi-GAN (V1), BigVGAN-base along with the ablation model without the filtered nonlinearity, and the high-capacity BigVGAN for the in-depth analysis.

The top row of the Figure 5 corresponds to a singing voice. The feature aliasing of HiFi-GAN introduces blurry harmonics because aliased features at multiple incorrect frequency components are aggregated during the generative process, which amplifies the error. BigVGAN-base without the filtered nonlinearity improves the harmonic components using the periodic activation and the advanced discriminator. BigVGAN models further improve the accuracy using the continuous feature representation and the anti-aliasing filter. The bottom row of the Figure 5 shows an instrumental audio. Similar to the singing voice example, HiFi-GAN exhibits blurry harmonics. BigVGAN models capture such challenging high-frequency harmonics significantly better than the baselines, which suggests that BigVGAN is robust to various unseen conditions. The visual analysis shows that BigVGAN exhibits a significantly less distortion of the harmonic components and a better sound quality for various types of audio beyond the clean speech signal.

## E ADDITIONAL RESULTS

In addition to the results in Section 4 of main text, we provide additional comparison with previous work and ablation models with the automatic objective evaluation.

**Comparison with Baselines Models** Table 7 and 8 show the expanded objective results evaluated on LibriTTS, including SC-WaveRNN (Paul et al., 2020), WaveGlow (Prenger et al., 2019), WaveFlow (Ping et al., 2020), UnivNet (Jang et al., 2021), and the ablation models of BigVGAN-base.

SC-WaveRNN (Paul et al., 2020) is a universal vocoder based on WaveRNN (Kalchbrenner et al., 2018) with a speaker embedding from a separately trained encoder network. We replicate the training procedure following (Paul et al., 2020) using the official implementation, while matching training data and audio hyperparameters used in this study (i.e., LibriTTS-`full`, 24,000Hz sampling rate and a 100-band log-mel spectrogram with [0, 12] kHz frequency). We train SC-WaveRNN for 3M steps. Although the model can generate speech with diverse speaker identities, we find that the performance is significantly worse than GAN vocoders both in objective scores and human listening test. It fails to generate intelligible audio for unseen recording environments and out-of-distribution sample. We conjecture that the speaker embedding network is not robust enough to generalize to such setups.

WaveGlow (Prenger et al., 2019) and WaveFlow (Ping et al., 2020) are two widely known vocoders based on normalizing flows (Rezende & Mohamed, 2015). We train the flow-based vocoders using the STFT hyperparameters used in this study. WaveGlow is trained for 2M steps and WaveFlow is trained for 1M steps as suggested by the authors. Flow-based vocoder features fast and parallel synthesis with an analytic likelihood objective due to its bijectivity. However, the quality of flow-based vocoders degrades heavily for multi-speaker setup. This renders the flow-based model unsuitable as a universal neural vocoder.

We also train UnivNet-c32 (Jang et al., 2021) on LibriTTS `train-full` using the open-source implementation (note that UnivNet uses the 24,000Hz sampling rate and the 100-band log-mel spectrogram with [0, 12] kHz frequency by default). However, unlike the HiFi-GAN and BigVGAN architectures, using the large-scale dataset did not improve the quality of the UnivNet architecture. Overall, the publicly available model trained on `train-clean-360` scores marginally better in terms of PESQ and periodicity error compared to our checkpoint trained on `train-full`. Other metrics exhibit different preference depending on the training set. The subjective quality is indistinguishable to the authors between the two checkpoints, including the out-of-distribution setup. We conjecture that the UnivNet architecture is harder to generalize to the unseen recording environments. For subjective SMOS tests, we choose to use the publicly available checkpoint using `train-clean-360` to faithfully represent the performance of the previous work. Note that the open-source implementation of UnivNet is unofficial. Therefore, our observation does not lead to the conclusion that UnivNet is worse than HiFi-GAN.

**Comparison with Ablation Models** From both clean and other recording environments, BigVGAN-base shows consistent improvements in all metrics compared to its ablation models. Specifically, BigVGAN-base (without filter & snake) refers to the vanilla HiFi-GAN architecture trained with MRD replacing MSD. Similar to the findings in (Jang et al., 2021), MRD provides more accurate modeling of the waveform by sharpening the spectral structure. Applying *Snake* activation (BigVGAN-base without filter) increases accuracy and reduces the periodicity error from the desired inductive bias. Finally, incorporating the continuous feature representation and the low-pass filter (BigVGAN-base) provides the best accuracy by suppressing aliasing and high frequency artifacts. Our final 112M BigVGAN substantially improves the accuracy for the state-of-the-art universal neural vocoding.

**Results on Unseen VCTK and LJSpeech Dataset** Table 9 and 10 show the objective speech evaluation metric results gathered from VCTK and LJSpeech dataset, consistent with the results from Table 7 and 8. Although the dataset is not included for training BigVGAN and the baseline HiFi-GAN, all GAN-based models performed comparatively well from our subjective listening test. This indicates that given diverse enough training data, modern non-autoregressive GAN vocoder can synthesize high-quality speech with unseen speaker identities from the clean recording environment.

Table 7: Objective results of BigVGAN from `dev-clean` of LibriTTS including ablation models of BigGAN-base and previous work. †: pretrained weight obtained from an open-source repository which used `train-clean-360` subset for training.

| LibriTTS (clean) | MAE(↓) | M-STFT(↓) | PESQ(↑) | MCD(↓) | Periodicity(↓) | V/UV F1(↑) |
|---|---|---|---|---|---|---|
| SC-WaveRNN | 0.5517 | 2.1411 | 1.774 | 1.5854 | 0.2925 | 0.8300 |
| WaveGlow-256 | 0.5368 | 1.3238 | 3.179 | 2.3897 | 0.1423 | 0.9419 |
| WaveFlow-128 | 0.2839 | 1.0706 | 3.120 | 0.9632 | 0.1339 | 0.9459 |
| UnivNet-c32† | 0.2803 | 0.9552 | 3.348 | 0.7017 | 0.1342 | 0.9433 |
| UnivNet-c32 | 0.2772 | 0.9433 | 3.310 | 0.6942 | 0.1356 | 0.9435 |
| HiFi-GAN (V1) | 0.2579 | 0.9773 | 3.042 | 0.6257 | 0.1545 | 0.9306 |
| BigVGAN-base | 0.1546 | 0.8569 | 3.574 | 0.4180 | 0.1298 | 0.9475 |
| w/o filter | 0.1770 | 0.8838 | 3.472 | 0.4624 | 0.1354 | 0.9437 |
| w/o filter & snake | 0.1899 | 0.9047 | 3.351 | 0.4852 | 0.1399 | 0.9401 |
| **BigVGAN** | **0.0931** | **0.7796** | **4.053** | **0.3392** | **0.1013** | **0.9610** |

Table 8: Objective results of BigVGAN from `dev-other` of LibriTTS including ablation models of BigGAN-base and previous work.

| LibriTTS (other) | MAE(↓) | M-STFT(↓) | PESQ(↑) | MCD(↓) | Periodicity(↓) | V/UV F1(↑) |
|---|---|---|---|---|---|---|
| SC-WaveRNN | 0.6125 | 2.3274 | 1.576 | 1.4717 | 0.2174 | 0.8896 |
| WaveGlow-256 | 0.5096 | 1.2960 | 3.098 | 2.3284 | 0.1546 | 0.9337 |
| WaveFlow-128 | 0.3359 | 1.1533 | 2.935 | 1.5278 | 0.1493 | 0.9360 |
| UnivNet-c32† | 0.3027 | 0.9973 | 3.166 | 0.8643 | 0.1439 | 0.9345 |
| UnivNet-c32 | 0.2998 | 0.9883 | 3.107 | 0.8495 | 0.1457 | 0.9337 |
| HiFi-GAN (V1) | 0.2724 | 1.026 | 2.853 | 0.6948 | 0.1585 | 0.9294 |
| BigVGAN-base | 0.1625 | 0.9006 | 3.464 | 0.4947 | 0.1276 | 0.9442 |
| w/o filter | 0.1844 | 0.9223 | 3.364 | 0.5017 | 0.1391 | 0.9395 |
| w/o filter & snake | 0.2008 | 0.9456 | 3.240 | 0.5389 | 0.1451 | 0.9344 |
| **BigVGAN** | **0.0986** | **0.8197** | **4.001** | **0.4097** | **0.1023** | **0.9586** |

Table 9: Objective results from unseen VCTK dataset. We used randomly selected 100 audio clips.

| VCTK | MAE(↓) | M-STFT(↓) | PESQ(↑) | MCD(↓) | Periodicity(↓) | V/UV F1(↑) |
|---|---|---|---|---|---|---|
| SC-WaveRNN | 0.6619 | 2.7344 | 1.445 | 1.7597 | 0.2528 | 0.8611 |
| WaveGlow-256 | 0.5454 | 1.3324 | 3.149 | 2.4512 | 0.1218 | 0.9513 |
| WaveFlow-128 | 0.2982 | 1.0825 | 2.953 | 1.1127 | 0.1213 | 0.9518 |
| UnivNet-c32† | 0.2753 | 0.9476 | 3.235 | 0.7180 | 0.1131 | 0.9535 |
| UnivNet-c32 | 0.2820 | 0.9586 | 3.184 | 0.7403 | 0.1198 | 0.9434 |
| HiFi-GAN (V1) | 0.2541 | 0.9859 | 3.029 | 0.6795 | 0.1336 | 0.9375 |
| BigVGAN-base | 0.1527 | 0.8677 | 3.443 | 0.4526 | 0.1047 | 0.9586 |
| w/o filter | 0.1739 | 0.8935 | 3.379 | 0.4924 | 0.1128 | 0.9528 |
| w/o filter & snake | 0.1885 | 0.9184 | 3.316 | 0.5159 | 0.1208 | 0.9481 |
| **BigVGAN** | **0.0925** | **0.7684** | **4.001** | **0.3557** | **0.0833** | **0.9672** |

Table 10: Objective results on unseen LJSpeech dataset. We used randomly selected 100 audio clips.

| LJSpeech | MAE(↓) | M-STFT(↓) | PESQ(↑) | MCD(↓) | Periodicity(↓) | V/UV F1(↑) |
|---|---|---|---|---|---|---|
| SC-WaveRNN | 1.0115 | 2.6994 | 1.233 | 4.8464 | 0.3907 | 0.7404 |
| WaveGlow-256 | 0.4933 | 1.2893 | 3.352 | 2.9921 | 0.1182 | 0.9561 |
| WaveFlow-128 | 0.3674 | 1.2402 | 3.072 | 2.7217 | 0.1170 | 0.9560 |
| UnivNet-c32† | 0.3418 | 1.0613 | 3.425 | 1.1903 | 0.1210 | 0.9519 |
| UnivNet-c32 | 0.3356 | 1.0429 | 3.384 | 1.1356 | 0.1230 | 0.9503 |
| HiFi-GAN (V1) | 0.3008 | 1.0950 | 3.210 | 1.7370 | 0.1347 | 0.9456 |
| BigVGAN-base | 0.1747 | 0.9121 | 3.741 | 0.8626 | 0.1164 | 0.9548 |
| w/o filter | 0.2015 | 0.9395 | 3.662 | 0.9715 | 0.1169 | 0.9548 |
| w/o filter & snake | 0.2263 | 0.9946 | 3.521 | 1.1320 | 0.1299 | 0.9479 |
| **BigVGAN** | **0.1102** | **0.8554** | **4.112** | **0.7164** | **0.0957** | **0.9642** |

Table 11: Character error rates (CER) of synthesized speech on multiple languages.

| Method | English | German | Catalan | Spanish | French | Mandarin |
|---|---|---|---|---|---|---|
| Ground Truth | 6.298 | 3.785 | 3.001 | 4.194 | 5.450 | 25.431 |
| SC-WaveRNN | 15.155 | 9.508 | 5.763 | 10.198 | 12.427 | 47.653 |
| UnivNet-c32† | 6.914 | 4.037 | 3.058 | 4.439 | 5.717 | 26.504 |
| HiFi-GAN (V1) | 6.906 | 4.070 | 3.082 | 4.490 | 5.770 | 27.179 |
| HiFi-GAN (112M) | 6.615 | 3.920 | 3.043 | 4.315 | 5.608 | 26.174 |
| BigGVAN-base | 6.574 | 3.905 | 3.034 | 4.335 | 5.594 | 26.174 |
| BigVGAN | **6.436** | **3.829** | **3.028** | **4.261** | **5.517** | **25.756** |

## F    LINGUISTIC ACCURACY EVALUATION

In this section, we evaluate the quality of neural vocoders in terms of linguistic accuracy. To measure linguistic accuracy of the synthesized speech from BigVGAN on in-distribution and various out-of-distribution languages, we conduct experiments using a high-performance Conformer-based (Gulati et al., 2020) automatic speech recognition (ASR) model on multiple languages. We use `conformer_transducer_large` model, where the pretrained checkpoints for the considered languages are publicly available from the NVIDIA NeMo (Kuchaiev et al., 2019) toolkit. We generate samples from BigVGAN using test set corpus from Mozilla Common Voice (MCV) 8.0 dataset for the following languages: English (`en`), German (`de`), Catalan (`ca`), Spanish (`es`), French (`fr`), and Mandarin (`zh`). Then, we obtain transcriptions from the synthesized speech using the ASR model and calculate character error rate (CER). In this way, we can assess a degree of degradation in linguistic accuracy compared to the CER of the ground truth speech.

The CER results are shown in Table 11. BigVGAN has the consistently lowest CERs, which are also close to ground truth. Furthermore, BigVGAN-base (14M) outperforms the largest HiFi-GAN (112M) on four (`en`, `de`, `ca`, `fr`) out of six languages, ties on one (`zh`), and underperforms only on one language (`es`). The results further validate that BigVGAN is the most linguistically accurate universal vocoder with the lowest artifacts and show the benefits of the model to be applied to various speech applications.

## G    ADDITIONAL DETAILS OF MECHANICAL TURK EVALUATION

We use Mechanical Turk for MOS and SMOS tests with 450 unique ratings per model. We allow each worker to evaluate up to three random samples for diverse participants. Crowdsourcing evaluation is considered to be a noisy process. We apply several filtering criteria to improve the reliability of workers : i) We restrict the listeners to be native English speakers in the United States. ii) We only allow listeners with the evaluation acceptance rate higher than 98%, and the number of previously approved HITs higher than 100 to participate in this test. iii) We conduct 5-scale MOS/SMOS evaluation with random ordering of the model samples, with the inclusion of ground-truth samples as the hidden control questions. We apply filtering by rejecting ratings from those workers who score the ground-truth samples as 3 or even lower, because the listeners are instructed to give score 5 for ground-truth samples after listening to the same ground-truth sample prior to the rating.

