# OpenReview forum: "BigVGAN: A Universal Neural Vocoder with Large-Scale Training"
_ICLR.cc/2023/Conference — ICLR 2023 poster_

### Official Review · Reviewer_qkXX · 2022-10-24

**Confidence:** 4
**Correctness:** 4
**Technical Novelty And Significance:** 3
**Empirical Novelty And Significance:** 3
**Recommendation:** 8

**Clarity, Quality, Novelty And Reproducibility:**

The paper is clear and well written. The novelty is mostly in scaling and in adopting some architectural choices that improve the scalability of the model. Paper is clear and well written.

**Details Of Ethics Concerns:**

Specifically the ability to generalize to unseen languages, will provide great ethic considerations to this model, because it will allow to address data-limited languages that may not decode correctly with previous models. This is a very positive point of scaling up the model and with this work in particular.

**Strength And Weaknesses:**

The main strength of this work is the scaling up, including using the whole Libri dataset, including different recording conditions. This shows up in increased robustness at generation time, as I mention below. The architecture is also well motivated. Also it is well known that scaling up GAN training is not easy due to instability in the process, as outlined by the authors.
The increased bandwidth is also a plus for the model.

Listening to the samples most of the practical improvement seems to come from samples with music / noise in the background. Also foreign languages seem to be better model. I personally could not hear a major difference from unseen speakers compared to baseline methods, but this is just my personal opinion.

**Summary Of The Paper:**

The paper addresses the limits of GAN based vocoders to generalize to new/different voices not seen in the training set.
It does this by scaling up the generator block of the model an using periodic activations. Specifically the paper call for using the SNAKE activation function, in combination with 2x upsampling and downsampling used to control high-frequency artifacts from SNAKE.



**Summary Of The Review:**

This is a good paper, with extensive well motivated experiments.
I would have guessed a bigger improvement from qualitative tests, especially for out of distribution, especially when I listen to the samples.
Some of the baseline models seem to be completely broken in the samples presented, however the difference in the scores is rather small.
I look forward for the release of the code and the models.

---

> ### Author Response · Authors · 2022-11-18
> **To Reviewer qkXX**
>
> Many thanks for your review, which is really helpful to improve the quality of our work. We will address your comments in the following.
>
> “Listening to the samples most of the practical improvement seems to come from samples with music / noise in the background. Also foreign languages seem to be better model. I personally could not hear a major difference from unseen speakers compared to baseline methods, but this is just my personal opinion.”
> * Yes, we agree that the improvement is relatively small between BigVGAN and the baseline methods on clean speech in general, although we think one exception is the low-frequency male voice. In our opinion (also subjective), BigVGAN provides solid improvement at capturing the low-frequency pitch of the male voice, while HiFi-GAN tends to produce inaccurate pitch. For example, one may check the pronunciation of “surroundings” in the 2nd example of LibriTTS clean. Note that the low-frequency male voices were believed to be more difficult to model among speech synthesis practitioners, because they were easily entangled with some unavoidable low-frequency noises (e.g., mouth noises).
>
> “I would have guessed a bigger improvement from qualitative tests, especially for out of distribution, especially when I listen to the samples.”
> * Thank you for mentioning this. In general, the subjective evaluation of universal vocoding is a challenging process. In particular, there is a noticeable gap between expert listeners (e.g., our workers, expert reviewers) and crowdsourced listeners. At a lot of times, we find big improvement according to expert listeners, but narrowed improvement of MOS scores according to random listeners. We think this comes down to the diversity of test samples in universal vocoding (e.g., various speakers, recording environments, unseen languages, audio genres). The listeners need to pay a lot of attention during evaluation, which is hard to enforce on crowdsourcing platform.

---

### Official Review · Reviewer_6Yoq · 2022-10-25

**Confidence:** 4
**Correctness:** 3
**Technical Novelty And Significance:** 2
**Empirical Novelty And Significance:** 3
**Recommendation:** 8

**Clarity, Quality, Novelty And Reproducibility:**

The paper is well written. Some mathematical details could have been included in the main text rather than being relegated to the appendices.

**Strength And Weaknesses:**

Strengths:
- The paper presents the proposed methods in an organized fashion, in comparison to the existing work. It is clear what's the new contribution and how they contribute to the performance of the model.
- The paper presents a rich set of large-scale experimental results on various test scenarios, including OOD cases (on unseen languages) and noise conditions.
- The paper validates the experimental results using various objective and subjective metrics, including a newly developed SMOS metric that is convincing for this particular use case.

Weaknesses:
- It is not very clear which part of the model is contributing to better performance. The model is equipped with newly introduced activation functions and the AMP module, which are all making sense in terms of improving objective quality, but not too relevant to the SMOS performance, I believe. Then, the true contribution of the paper in terms of generalization power is just the model's capability of learning from the bigger LibriTTS dataset (involving all the noisy folds). It is a valid contribution point, but I would say then it's just "beating the state-of-the-art with a bigger computing power" situation, which is scientifically not significant. For example, I wonder what happens if HiFi-GAN was trained from the same large training set.
- The MOS and SMOS tests were done via crowdsourcing, but it's not mentioned if the listeners are at least native English speakers for the LibriTTS experiments. Given that MOS on AMT isn't that reliable, the paper could discuss more on their filtering criteria and so on.
- The snake activation seems to work well for the proposed vocoder, but it's a mere adaptation of existing work.

**Summary Of The Paper:**

The paper proposes a neural vocoder based on improved GAN architecture and the use of a bigger training dataset. The system aims at a "universal" vocoding function that works on various unseen categories of signals, including different acoustic signatures and languages.  The main claims are the proposed GAN-based approach is faster and more parallelizable than existing autoregressive models while it provides more flexible architectural choices than flow-based models. The main contribution of the paper is the introduction of the "snake" activation and the anti-aliasing module, that empirically improved the synthesis quality.

**Summary Of The Review:**

The paper presents a novel GAN-based universal vocoder. While the paper is not entirely clear about which part of the model is the true contributor to its better performance, especially given that it's trained from a much larger dataset, the paper did improve the state-of-the-art. The paper validated the synthesis results thoroughly and the proposed model appears to generalize better to the unseen test environments than the baseline model.

---

> ### Author Response · Authors · 2022-11-18
> **To Reviewer 6Yoq**
>
> Thank you so much for your detailed comments. We will address your comments in the following.
>
> “It is not very clear which part of the model is contributing to better performance. The model is equipped with newly introduced activation functions and the AMP module, which are all making sense in terms of improving objective quality, but not too relevant to the SMOS performance, I believe. Then, the true contribution of the paper in terms of generalization power is just the model's capability of learning from the bigger LibriTTS dataset (involving all the noisy folds). It is a valid contribution point, but I would say then it's just "beating the state-of-the-art with a bigger computing power" situation, which is scientifically not significant. For example, I wonder what happens if HiFi-GAN was trained from the same large training set.”
> * Thank you for raising the question. To be clear, we train all baseline models, including HiFi-GAN, on the full LibriTTS dataset (involving train-clean-100 + train-clean-360 + train-other-500). Thus, BigVGAN outperforms HiFi-GAN if it was trained on the same large training set. We further clarify this at the beginning of the Results section. In addition, the ablation study of model architecture by removing snake nonlinearity and anti-aliasing filter can be found in Table 4. The result demonstrates that both snake activation and anti-aliasing module contribute to the final good performance. The ablation study of using less diverse training data can be found in Table 5. The result demonstrates that a large training set is helpful too.
>
> “The MOS and SMOS tests were done via crowdsourcing, but it's not mentioned if the listeners are at least native English speakers for the LibriTTS experiments. Given that MOS on AMT isn't that reliable, the paper could discuss more on their filtering criteria and so on.”
> * This is a great question. Since crowdsourcing evaluation is noisy, we indeed apply several filtering criteria to improve the reliability of workers : i) We restrict the listeners to be native English speakers in the United States. ii) We only allow listeners with the evaluation acceptance rate higher than 98%, and the number of previously approved ratings higher than 100 to participate in this test. iii) We conduct 5-scale MOS/SMOS evaluation with random ordering of the model samples, with the inclusion of ground-truth samples as the hidden control questions. We apply filtering by rejecting ratings from those workers who rated the ground-truth  sample as 3 or even lower. We have included a discussion of these filtering criteria in Appendix G of updated draft.
>
> “The snake activation seems to work well for the proposed vocoder, but it's a mere adaptation of existing work.”
> * We agree the snake activation is an adaptation of existing work, but we think this adaptation is non-trivial and constitutes a significant contribution for the audio / speech synthesis community. First of all, although snake activation is demonstrated to be effective for temperature and financial data prediction in the original work [Liu+2020], it is unknown that it can work well for large-scale generative modeling of raw audio, which is more challenging in terms of the scale and sequence length (i.e., 1s audio waveform corresponds to 24,000 samples). More importantly, we propose a novel multi-periodicity composition module adding features from multiple residual blocks with different channel-wise learnable periodicities for snake activation, which is the key for advancing the state-of-the-art result in the field of  audio synthesis.

---

### Official Review · Reviewer_iGrd · 2022-10-26

**Confidence:** 5
**Correctness:** 3
**Technical Novelty And Significance:** 3
**Empirical Novelty And Significance:** 4
**Recommendation:** 8

**Clarity, Quality, Novelty And Reproducibility:**

**Clarity**

Overall, this paper is well-written and easy to read. The authors provide sufficient detail to understand both their methodological decisions and extensive experimentation. A few low-level comments on clarity:

- (Introduction) “where the studio-quality recordings are not available” to which of the many examples in the list does this dangling clause belong?
- (Introduction) “This architectural flexibility leads to large model capacity when we scale up the size of the model.” This argument doesn’t make sense to me. One could definitely scale up low-based models arbitrarily (though whether or not they train effectively is another matter)
- (Section 3.4) “.. by scaling up the generator’s model size to its maximum ..” why does the current configuration constitute a maximum?

**Quality**

The methodology in this paper is mostly bits and pieces of other papers cobbled together, and parts of these descriptions unfortunately come across as alchemy. However, the modifications are all properly ablated, and the high-level arguments suggested by the authors do make sense intuitively. A few questions about the methodology:

- (Figure 1) In the AMP block, surely the second low-pass filter is applied _before_ downsample1d? Otherwise, if the figure is correct, downsample1d will alias since you haven’t low-passed yet
- (Section 3.3) The upsampling/downsampling procedure does _reduce_ aliasing risk, but it doesn’t appear to _eliminate_ it. Why not just actually band limit the alpha parameter using a sigmoid or something similar?

There is a missing citation in the first paragraph to WaveGAN from Donahue et al. 2019, an earlier ICLR paper which was the first non-autoregressive deep generative model capable of producing raw audio.

Another question I have relates to the homogeneity of the training data. If the goal is to achieve universal vocoding, why not train on a ton of non-speech data as well? E.g., the authors could consider training on AudioSet

**Novelty**: The methodological innovations and the strength of the experimental results (particularly out-of-domain) do constitute substantial novelty in my view.

**Reproducibility**: As is common in deep learning literature, it would be challenging if not impossible to reproduce the results of this paper from the descriptions alone. Fortunately, the authors indicate they will share code and pre-trained models upon publication.


**Strength And Weaknesses:**

Neural vocoding is a common component of many audio synthesis systems, and _universal_ vocoding has long been an enticing prospect for audio research. The qualitative performance of BigVGAN both in- and out-of-domain represents a tangible step towards universal vocoding.

One potential limitation is that BigVGAN outputs 24k audio, which is lower than the standards used for high-fidelity music audio. Another limitation is that GAN training remains fickle---a lot of the architectural modifications and hyperparameter adjustments the authors may be fragile and difficult for subsequent researchers to build on.

**Summary Of The Paper:**

This paper presents BigVGAN, a “universal” neural vocoder which achieves impressive results both in- and out-of-domain. The authors detail extensive architectural modifications and hyperparameter tuning necessary to train this model. Thorough comparisons and ablations are conducted to justify the proposed method and demonstrate its efficacy as a universal vocoder.

**Summary Of The Review:**

Overall, BigVGAN represents an exciting step towards universal vocoding that will certainly be of interest to many audio researchers, and will likely become a common component of many audio generation systems. I believe it could be an important inclusion in this year’s ICLR conference proceedings.

---

> ### Author Response · Authors · 2022-11-18
> **To Reviewer iGrd -- Part 1**
>
> Thank you so much for your review. They are very helpful to improve the quality of our paper. We will address your comments in the following.
>
> 1, “One potential limitation is that BigVGAN outputs 24k audio, which is lower than the standards used for high-fidelity music audio.”
> * This is a great suggestion. We will train BigVGAN on 44k or 48kHz music audio as an interesting future work.
>
> 2, “Another limitation is that GAN training remains fickle---a lot of the architectural modifications and hyperparameter adjustments from the authors may be fragile and difficult for subsequent researchers to build on.”
> * To facilitate future research, we provide the detailed recipe in Section 3.4 and practical lessons for large-scale GAN training in Appendix C. We will release the code and models for subsequent researchers to build on. So far, we find the current architecture and hyperparameters are robust for different training data.
>
> 3, “(Introduction) “where the studio-quality recordings are not available” to which of the many examples in the list does this dangling clause belong?”
> * Thanks for raising this question. For clarity, we have removed this dangling clause in the updated draft and added a subsequent sentence: “In these applications, the neural vocoder also needs to generalize well for audio recorded at various conditions.”
>
> 4, “(Introduction) “This architectural flexibility leads to large model capacity when we scale up the size of the model.” This argument doesn’t make sense to me. One could definitely scale up flow-based models arbitrarily (though whether or not they train effectively is another matter)”
> * This sentence was meant for a comparison between GAN and non-autoregressive flow-based models. In non-autoregressive flow-based models (Glow, WaveGlow), the affine coupling layer was introduced for maintaining the bijection between latent and data, which leads to relatively small model capacity compared to autoregressive models given the same number of parameters. WaveFlow paper provides a detailed analysis about this observation. In contrast, the GAN architecture does not have such bijection constraints. For clarity, we have rephrased the sentence in the updated draft. Thanks again for your comment.
>
> 5, “(Section 3.4) “.. by scaling up the generator’s model size to its maximum ..” why does the current configuration constitute a maximum?”
> * Thank you for mentioning this. In our previous endeavor, models larger than 112M failed to maintain training stability. The potential reason could be the suboptimal architecture configurations of larger models. It is interesting to further scale up the model in future work. We have rephrased the sentence in the revision.
>
> 6, “(Figure 1) In the AMP block, surely the second low-pass filter is applied before downsample1d? Otherwise, if the figure is correct, downsample1d will alias since you haven’t low-passed yet”
> * This is a good question. Technically speaking, the low-pass filter (Kaiser sinc) is applied as the kernel in conv1d for downsampling, and as the kernel in transposed conv1d for upsampling. So, the low-pass filter and downsampling are essentially performed at the same time. We will clarify it in the draft and release the implementation code. The method is similar to the one shown in the StyleGAN3 paper, and the downsampling can still alias the feature.  Similar to the SytleGAN3, we found that this 2x up/down sampling along with low-pass filters was good enough to improve the quality.
>
> 7, “(Section 3.3) The upsampling/downsampling procedure does reduce aliasing risk, but it doesn’t appear to eliminate it. Why not just actually band limit the alpha parameter using a sigmoid or something similar?”
> * Thank you for the suggestion on band limiting alpha parameters. We think this direction is interesting. To implement that, one needs to carefully investigate on how to band limit alphas for given different feature resolutions after each transposed conv1D. We would like to investigate this in future work.

---

> > ### Author Response · Authors · 2022-11-18
> > **To Reviewer iGrd -- Part 2**
> >
> > 8, “There is a missing citation in the first paragraph to WaveGAN from Donahue et al. 2019, an earlier ICLR paper which was the first non-autoregressive deep generative model capable of producing raw audio.”
> > * Thank you for reminding us of this missing citation. We have included it in the first paragraph in the updated draft.
> >
> > 9, “Another question I have relates to the homogeneity of the training data. If the goal is to achieve universal vocoding, why not train on a ton of non-speech data as well? E.g., the authors could consider training on AudioSet”
> > * This is a nice suggestion. In this work, we improve the out-of-distribution generalization of GAN vocoder by optimizing its architecture and scaling up its model size. We also investigate the impact of training data diversity in ablation study (i.e., LibriTTS train-full vs. train-clean-360 vs. VCTK). It is very interesting to scale our work in terms of training data beyond speech, such as AudioSet as you suggested. One may expect better performance on non-speech data, if the model is trained on non-speech data. However, the zero-shot performance on OOD test data is still quite important, as it is challenging (if not impossible) to cover all real-world sounds within training data.

---

### Official Review · Reviewer_F87e · 2022-10-27

**Confidence:** 4
**Correctness:** 4
**Technical Novelty And Significance:** 3
**Empirical Novelty And Significance:** 3
**Recommendation:** 8

**Clarity, Quality, Novelty And Reproducibility:**

The paper is very well written and easy to understand and follow. The ideas in the paper are novel and is widely applicable for audio generation tasks. Enough details have been provided to enable reproducibility.

**Strength And Weaknesses:**

Strengths:
* Addresses some of the key challenges in raw audio synthesis
* Strong experiments and results


**Summary Of The Paper:**

The paper presents two key ideas: periodic activation function and anti-aliased activation. Both of these choices enable the generation of raw audio from mel-spectrogram with unprecedented quality for unseen speakers and recording conditions.

**Summary Of The Review:**

The paper addresses some of the key challenges plaguing GAN-based audio waveform synthesis. Experiments justify the claims.

---

> ### Author Response · Authors · 2022-11-18
> **To Reviewer F87e**
>
> Thank you so much for your review. The code and model will be released later. Hope this could facilitate future endeavors of audio synthesis.

---

### Official Review · Reviewer_ozMy · 2022-10-31

**Confidence:** 3
**Correctness:** 3
**Technical Novelty And Significance:** 2
**Empirical Novelty And Significance:** 3
**Recommendation:** 6

**Clarity, Quality, Novelty And Reproducibility:**

The paper is very clearly written with all the relevant information needed to evaluate the work except the code and model-specific details that are still awaited. The webpage with extensive sound examples is really helpful in evaluating the incremental advantage of components of BigVGAN and its comparison with previous models. The figures are clear and adequately captioned. More information on the model would have been useful but overall, it is a high quality paper that explains its result clearly and answers a lot of the questions a reader might have in the appendix section which is quite elaborate.

For reproducibility, the authors have promised model details and full code.


**Strength And Weaknesses:**

Strengths:
- The model is able to outperform all the previous state-of-the art neural synthesis tools even with a scaled down version
- The zero shot performance on OOD results is impressive
- The periodic inductive bias on activations is a very useful addition for more structured types of sounds like speech and music
- The authors were able to train a very large GAN for audio synthesis without some of the commonly encountered training issues specific to this setting

Weaknesses-
- Although the authors use a larger frequency range than previous papers, it still falls short of the claimed ‘full-band’ especially for non-speech sounds where there could be useful information contained in the frequency bands above 12 kHz
- The SMOS and MOS results are not able to show statistically significant improvement in BigVGAN vs. HiFi-GAN and BigVGAN-base vs. BigVGAN
- The periodic inductive bias doesn’t make a lot of theoretical sense for sound textures and other physics-based sounds
- Outside the training set doesn’t necessary mean out of distribution



**Summary Of The Paper:**

The paper presents BigVGAN, a GAN based vocoder that generalizes well for OOD scenarios without additional finetuning. They seem to achieve state-of-the-art performance for audio synthesis in a variety of novel scenarios like new speakers, unseen recording environments etc. The proposed model is able to reproduce a lot of fine details across sound textures, speech and music that previous state-of-the-art models were not able to replicate. The authors show that introducing periodic activations into the generator provides the desired inductive bias for audio synthesis.

**Summary Of The Review:**

Overall, this is a well-written manuscript that presents research that pushes the state-of-the-art in neural audio synthesis using GANs. BigVGAN generalises fairly outside the training set, better than previous models on all major types of structured sounds. The use of a periodic inductive bias through activations is a smart theoretical idea and a choice that seems to work well based on the full model and lesion results. That said, the work has minor limitations including but not limited to lack of statistical analysis for mturk results that seem to be statistically insignificant for some vital comparisons.

---

> ### Author Response · Authors · 2022-11-18
> **To Reviewer ozMy**
>
> Thank you so much for your review. We will address your comments in the following.
>
> 1, “Although the authors use a larger frequency range than previous papers, it still falls short of the claimed ‘full-band’ especially for non-speech sounds where there could be useful information contained in the frequency bands above 12 kHz.”
> * Many thanks for your comment. Because our model is trained on 24kHz audio, we used the term “full-band” to describe the conditioning 100 band mel-spectrogram with frequency range [0, 12kHz] based on the Nyquist-Shannon sampling theorem. In contrast, conventional mel-spectrograms are engineered to have a limited frequency band [0, 8kHz] by cutting off the high frequency details for easier modeling. Since the term “full-band” raises confusion, we have removed it in the updated draft. We will train BigVGAN on 44kHz or 48kHz audio as future work.
>
> 2, “The SMOS and MOS results are not able to show statistically significant improvement in BigVGAN vs. HiFi-GAN and BigVGAN-base vs. BigVGAN.” AND “That said, the work has minor limitations including but not limited to lack of statistical analysis for mturk results that seem to be statistically insignificant for some vital comparisons.”
> * For most of out-of-distribution tests, including noisy recordings of unseen languages in Table 3 and music data (i.e. MUSDB18-HQ) in Table 4, we observed statistical significant improvement in BigVGAN vs. HiFi-GAN. For example, even BigVGAN-base shows statistically significant improvements compared to the baseline HiFi-GAN with p-value < 0.05 from the Wilcoxon signed-rank test for Noisy (sim) and Noisy (real) categories. For Others and Mixture categories of MUSDB18-HQ,  one could observe noticeable improvement in BigVGAN vs. BigVGAN-base. That said, we recognize that BigVGAN models have limited improvements for clean speech data (e.g. LibriTTS test), partially because baseline models (e.g. HiFi-GAN) already perform perceptually well for clean speech.
>
> 3, “The periodic inductive bias doesn’t make a lot of theoretical sense for sound textures and other physics-based sounds”. AND “The use of a periodic inductive bias through activations is a smart theoretical idea and a choice that seems to work well based on the full model and lesion results.”
> * This is a good question. There could be situations where the sound signal doesn't naturally contain periodic components. Despite this, the signal can still be well approximated by the summation of periodic functions because of the Fourier series / decomposition. Empirically, we found the periodic nonlinearities provide better extrapolation capability in various tests.
>
> 4, “Outside the training set doesn’t necessary mean out of distribution.”
> * We agree on this point and appreciate your suggestion to clarify the definition of out of distribution in the updated manuscript. In our experiment, some of test data are out of distribution, not because they are outside the training set as in the classical train/test split, but because they are different types of audio data (e.g., unseen languages, unseen singing voices, music instrumental sound) compared to the training data (i.e. LibriTTS, an English speech dataset based on public domain audio books).

---

> > ### Comment · Reviewer_ozMy · 2022-12-04
> > **Thanks to the authors**
> >
> > Hi authors,
> >
> > Thanks for your detailed response and making suitable changes to the manuscript. I appreciate the time you took to think about the suggestions and answering concerns point-wise. Although, I still think that the approach is not generally useful for a vast class of sounds that are stochastic in nature unless they are tested as fourier decomposition for such sounds doesn't always help. That said, this work is definitely a notable improvement over previous GAN-based vocoders and should be presented at the conference.

---

> > > ### Author Response · Authors · 2022-12-08
> > > **Many thanks for your reply**
> > >
> > > Dear reviewer,
> > >
> > > Many thanks for taking time to read our response and reply nicely. We also appreciate your valuable suggestions during this review process.
> > >
> > > We want to mention that natural sounds in the real world  (e.g., speech, instrumental audio, music) can be represented as a mixture of their component Sinusoidal waves of different frequencies, because all sound waves essentially represent the vibrations of medium back and forth. Our approach is based on this very general assumption. We test our approach for modeling these natural sounds and demonstrate notable improvement over previous methods. Note that, BigVGAN is also good at modeling various types of noises, including purely stochastic white noise, as we demonstrated in several noisy speech experiments. We will look for the potential limitation of periodic inductive bias in future work.

---

### Decision · Program_Chairs · 2023-01-20

**Decision:**

Accept: poster

**Justification For Why Not Higher Score:**

The work is more like engineering efforts. The scientific contributions and insight seem a little bit weak.

**Justification For Why Not Lower Score:**

Strong results.

**Metareview: Summary, Strengths And Weaknesses:**

Four experts reviewed this paper with all accepted recommendations. The area chairs agree that this work makes a very important contribution by introducing a novel GAN-based universal vocoder and significantly improving the  SOTA.  The reviewers did raise some valuable concerns that should be addressed in the final camera-ready version of the paper. The authors are encouraged to make the necessary changes and include the missing references in the final version.



**Note From Pc:**

if the above contains the word "oral" or "spotlight" please see: "oral" presentation means -> notable-top-5% and "spotlight" means -> notable-top-25%. As stated in our emails, we are disassociating presentation type from AC recommendations